# Variance in odds ratios for estimating the deterrent effect of darkness on cycling: Variation due to the choice of case and control hours

**Steve Fotios\*, Jim Uttley, Demet Yesiltepe, Maan Balela**

Sheffield School of Architecture, University of Sheffield, Sheffield, United Kingdom

\* steve.fotios@sheffield.ac.uk

**Data Availability Statement:** All data are in the manuscript and/or supporting information files.

**Funding:** This work was conducted within SATURN (Supporting Active Travel Using Road-lighting at

## Abstract

Comparing the counts of cyclists travelling at specific times of day is one approach to measuring the impact of ambient light level. Here we investigate one source of variance in the magnitude of change seen in previous research—the choice of case and control hour. This was done through an analysis of cyclist flows using data from multiple automated cyclist counters in five cities (Arlington, Bergen, Berlin, Birmingham and Leeds) to determine the odds ratios (OR) for each combination of case and control hour. The results tend to reveal odds ratios above 1.0 indicating that cycling can be deterred by darkness. The odds ratios varied with the choice of case and control hour. For two cities (Birmingham and Leeds), the impact was small, with little difference in ORs between any case and control hour combination. For three cities the variance in ORs was larger. To represent the impact of darkness on cycling flows across the range of case and control hours we suggest the Mantel-Haenszel pooled odds ratio is used, in which the odds ratio for each combination of case and control hour is weighted by the total number of cyclists in that combination. This suggested a statistically significant (p<0.001) deterrent effect of darkness in all five cities.

## 1. Introduction

When cyclists are asked what changes would support cycling, enhanced road lighting is found to be a significant factor for cycling after dark [1, 2]. Stated preferences are easily influenced by experimental design, for example by 'enhanced road lighting' being one of the options they are requested to consider, and thus being directly influenced by the experimenter's expectation of what might matter. An alternative approach to establishing whether changes in lighting support cycling is to observe behaviour, a revealed preference, and one possible measure is the numbers of people choosing to cycle. A first tentative study has shown that the deterrence of darkness is mitigated by road lighting, with higher light levels offering greater mitigation [3]. The basis of that study was to compare cyclist counts under different phases of ambient light: we report here further analyses conducted to test the robustness of that analysis.

Night), funded by the Engineering and Physical Sciences Research Council (EP/V043587/1). The funders had no role in study design, data collection and analysis, decision to publish, or preparation of the manuscript.

**Competing interests:** The authors have declared that no competing interests exist.

Phases of ambient light range from daylight, through twilight (civil, nautical, and astronomical) to darkness. These phases are characterised quantitatively by the altitude of the sun relative to the horizon and qualitatively by expectations of visibility [4], see Table 1.

Research of road lighting has investigated the effect of differences in ambient light level, typically between daylight and darkness, on outcomes such as road traffic collisions (RTC) [5–11], crime [12–15] and traffic flow [16–18]. An intention of these studies is to capture the numbers of events occurring at different ambient light levels but at the same time of day, thus isolating the effect of ambient light level from other time of day influences. Two approaches have been used to do this.

One approach is to take advantage of the biannual daylight savings clock change that occurs in many countries, where clocks are put forward by one hour at the start of daylight saving time (in spring) and back by one hour at the end of daylight saving time (in autumn or fall). For specific time windows in the morning and evening, this leads to a rapid transition from daylight to darkness (or vice versa) from the week before to the week after the clock change. The number of weeks either side of the clock change varies from one [9, 10, 12] to two [5, 8, 17] or more [6, 19]. While the inclusion of only one or a few weeks before and after the clock change leads to small data samples, that short period better ensures little other change, such as change in weather.

The second approach is to consider seasonal variation in ambient light level over the whole year, picking a time window in the morning or evening which is in daylight for one part of the year but in darkness for another part. This time window, of daylight and darkness at different times of the year, is the *Case hour*. It is an 'hour' because that is the smallest interval at which the secondary data used in these studies tends to be reported. Shorter case intervals would be better because they improve the researcher's ability to discriminate between phases of ambient light level. Where data are reported at specific times rather than in a time window this allows a more precise allocation into different case periods [11] but with the knowledge that there will be inaccuracies in the reporting of the time at which events occurred [16].

With the whole year method, in the northern hemisphere, the case hour is likely to be in daylight for the months of April to August and in darkness for the months October to February. Portions of the remaining months may be omitted to exclude events occurring in civil twilight where the benefit of ambient light for visual tasks is ambiguous. The whole year approach thus tends to enable a larger data sample to be captured than does the clock change approach, but that depends on the extent to which twilight is excluded and the number of weeks considered before and after a clock change.

**Table 1. Definitions of different ambient light conditions.**

| Ambient light condition | Solar altitude ($\theta_s$) | Visibility conditions* | Descriptive illuminance (lx)* |
|---|---|---|---|
| Daylight | $\theta_s > 0°$ | Illumination is very good | Horizontal surface under a cloudless sky: sun at horizon = 355 lx, sun at zenith = 103,000 lx, |
| Civil twilight | $0° > \theta_s > -6°$ | Enough illuminance exists to enable outdoor civil activity to continue unhindered without resorting to the use of electric street lighting | End of civil twilight: 4.3 lx |
| Nautical twilight | $-6° > \theta_s > -12°$ | The limit of the visibility of ships approaching a harbour | End of astronomical twilight: 0.001 lx |
| Astronomical twilight | $-12° > \theta_s > -18°$ | The instance of the last stage of receipt of light emanating from the sun | - |
| Night | $-18° \geq \theta_s$ | - | - |

*Definitions from Muneer [4].

**Table 2. Past studies using odds ratios to compare travel counts for pedestrians and cyclists in different ambient light levels.**

| Study | Data | | | Method of analysis | | | Road user | OR (95%CI) for effect of darkness on travel count* | Effect size** |
|---|---|---|---|---|---|---|---|---|---|
| | Location and period | Reported count interval | Period analysed | Case hour | Control hours | | | | |
| Uttley & Fotios 2017 | Arlington, Virginia, USA 2011–2016 | 15 min | Clock change | Spring: 18:00–18:59 Autumn: 17:00–17:59 | Spring: 16:30–17:29; 19:30–20:29; 14:30–15:29; 21:30–22:29 Autumn: 15:30–16:29; 18:30–19:29; 13:30–14:29; 20:30–21:29 | | Pedestrians | 1.62 (1.60–1.63) p<0.001 | Small |
| | | | | | | | Cyclists | 1.38 (1.37–1.39) p<0.001 | Small |
| Fotios, Uttley & Fox 2019 | Arlington, Virginia, USA 2012–2015 | 15 min | Whole year | 18:00–18:59 | 15:00–15:59; 21:00–21:59 | | Pedestrians | 1.93 (1.92–1.95) p<0.001 | Medium |
| | | | | | | | Cyclists | 1.67 (1.66–1.68) p<0.001 | Small |
| Uttley, Fotios & Lovelace 2020 | Birmingham, UK 2012–2015 | 60 min | Whole year | 18:00–18:59 | 14:00–14:59; 22:00–22:59 | | Cyclists | 1.32 (1.31–1.33) p<0.001 | Small |
| Fotios & Robbins 2022 | Cambridge, UK 2019–2020 | 60 min | Clock change | Spring: 18:00–18:59 Autumn: 17:00–17:59 | Spring: 14:00–14:59, 21:00–21:59 Autumn: 14:00–14:59, 21:00–21:59 | | Pedestrians | 1.29 (1.26–1.33) p<0.001 | Small |
| | | | | | | | Cyclists | 1.57 (1.52–1.62) p<0.001 | Small |
| Uttley, Fotios, Robbins, Moscoso 2023 | Bergen, Lillestrøm, Oslo, Kristiansand and Trondheim, Norway | 60 min | Clock change | Spring: 19:00–19:59 Autumn: 17:00–17:59 | Spring: 14:00–14:59, 21:00–21:59 Autumn: 14:00–14:59, 21:00–21:59 | | Cyclists | 1.13 (1.10–1.16) p<0.001 | Negligible |
| | | | Whole year | 18:00–18:59 | 13:00–13:59, 22:00–22:59 | | Cyclists | 1.05 (1.03–1.06) p<0.001 | Negligible |

*An OR>1.0 indicates a reduction in the numbers of road users after dark compared with the same period when in daylight

** Odds ratio effect size thresholds suggested by Olivier & Bell [27]

The ratio of the numbers of events (RTCs, crimes, or road users) in darkness to the number in daylight shows the influence of ambient light level. There are, however, factors other than ambient light level which change between the daylight and dark periods of successive weeks, such as changes in weather [20]. To account for this the numbers of events in control periods are also considered. *Control hours* are those which remain consistently daylit (or dark) on both sides of the clock change, or throughout the whole year, rather than changing between daylight and darkness. The ratio of events in the daylight and dark periods of the case hour is compared against the ratio of events in the same periods but in the control hour to produce an odds ratio (OR) [21]. This gives a measure of the effect of darkness on the occurrence of the events being measured. The further the OR departs from unity, the greater the effect of darkness.

This paper focuses on traffic flow, and in particular the flow of cyclists. Table 2 shows previous studies which have investigated the effect of ambient light level on pedestrian and cyclist flows using an OR. These studies have found that, for a specific time of day, there were fewer pedestrians and cyclists after dark [3, 17, 22–24], but did not find that change in ambient light level affected the numbers of motorised vehicles [16], supporting evidence that ambient light level influences the choice of travel mode [25].

While in each of the cases shown in Table 2 the difference of the OR from 1.0 is found to be statistically significant, the effect size varies from negligible to medium. To interpret effect sizes for analyses using ORs we use the thresholds of 1.22, 1.86 and 3.00 for small, medium, and large effect sizes [26, 27] with the assumption that these represent the minimum threshold for the stated effect size. The estimated ORs for cyclist flows vary from 1.05, indicating a negligible effect, to 1.67, indicating a small effect; for pedestrians they vary from 1.29 to 1.93, indicating effects of small and medium practical significance.

There are a number of differences between these studies, including the locations of the counters (and hence factors such as the weather, infrastructure, and cycling/walking culture), the reported count interval, and the method of analysis. In the current article we consider one explanation for the variation in odds ratios—the choice of case and control hours. This study is the first to examine whether the researchers' choice of case and control hours affects the outcome of analyses investigating the impact of ambient light levels on traffic flow. We test two hypotheses: that darkness deters cycling, as indicated by ORs >1.0, and that varying the choice of case and control hours will lead to different ORs.

Previous studies of cyclist flows have tended to use just one case hour. For the clock change method, the short time interval means there is only one choice of case hour, and furthermore a duration of less than one hour is desirable to enable twilight to be excluded [11]. For the whole year method, however, there tends to be more than one possible choice of case hour, the difference between these options being the proportion of the year for which the case hour is in daylight or darkness.

Previous studies have tended to use two control hours–one in daylight (the period between morning sunrise and evening sunset) and one in darkness (the period starting at the end of evening civil twilight and ending at the start of morning civil twilight). There are, however, many more options. For example, consider analyses using the whole year approach. For Birmingham, UK (latitude 52˚28'53.11" N) there are four possible evening case hours (from 17:00 to 20:59), one morning case hour (06:00–06:59) ten possible control hours (09:00 to 14:59 and 23:00 to 02:59), while in Arlington, VA, USA (38˚52'51.64"N) there are two possible evening case hours (18:00 to 19:59), no morning case hours, and fifteen possible control hours (08:00 to 15:59 and 22:00 to 04:59). Note that in our analysis we excluded case hours where the daylight or darkness period for that case hour was less than two months per year, targeting a larger sample and hence a more accurate estimate of the OR. The choice of which case and control hours to use and other inclusion criteria represent two of many researcher degrees of freedom; if different choices are made, different conclusions may be reached [28, 29].

In an analysis of cyclist and pedestrian flows in Arlington using the clock change method, Uttley and Fotios [17] used four control hours, two in daylight and two in darkness, these beginning either 1.5 hours or 3.5 hours away from the start of the case hour. The ORs determined for cyclists for each control hour varied from 1.36 to 1.43, and from 1.17 to 1.75 for pedestrians: one apparent trend is that the OR was closer to 1.0 for the control hours closer to the case hour (dark control and day control rather than late dark and early day: see Fig 3 in that paper [17]). This difference in OR contributes to the apparent difference in ORs reported by studies using different choices of control hours. When analysing count data for the same location but using instead the whole year method with two control hours, Fotios et al [22] found ORs of 1.75 and 1.22 for the control hours starting at 15:00 (daylight) and 21:00 (dark) respectively, revealing a much larger effect of control hour choice.

The choice of case and control hours may influence the OR because the numbers and/or the type of traveller (or purpose for travelling) may change significantly between certain hours of the day. We distinguish here between utilitarian and recreational cycling journeys following Wessel [30]. The distinction is made because these types of journey tend to be differently distributed throughout the day [31] and because cyclists on utilitarian journeys may be less likely to be deterred by darkness than are cyclists on recreational journeys, similar to the reported effects of weather [32, 33].

To investigate the impact of the choice of case and control hours, we compared the ORs obtained for cyclist flows for all combinations of case and control hour. This was done for counts of cyclists recorded using automated counters in five cities, three being the focus of previous studies (Arlington, USA; Bergen, Norway; and Birmingham, UK see Table 2), with two

further locations added (Leeds, UK and Berlin, Germany). This article extends an interim report on this work [34] through the inclusion of three further cities, the inclusion of morning case hours, and a revised method for establishing the weighted mean.

## 2 Method

ORs describing the impact of darkness on cyclist numbers were determined for five cities, Arlington, Bergen, Berlin, Birmingham and Leeds, using the same data sources (where relevant) and the same method of analysis (the whole year approach) as were used in previous analyses of cycling in those locations [3, 22, 24]. Note that the previously published analysis of cities in Norway included Lillestrøm, Oslo, Kristiansand and Trondheim in addition to Bergen [24]: we now use only Bergen as access to data for a larger number of counters was subsequently obtained. Note also that the previous analysis [24] defined 22:00–22:59 as a control hour for Norway to enable a consistent choice for all five cities: for Bergen it would be defined as a case hour by consideration of solar altitude, but was not used as such in the current work as it offers less than two months duration in either daylight or darkness (see below for inclusion criteria).

### 2.1 Data sources

This analysis used the numbers of cyclists passing a specific location as recorded by automated counters installed by others such as local authorities or organisations seeking to monitor traffic flow. Other than for Arlington the data are available only at hourly intervals, and hence a one-hour time window was used for all locations. For all five cities, the data were available to the public. Table 3 shows the data sources, the numbers of counters at each location and the ranges for which data were analysed.

The current analysis used, as closely as possible, the same samples as were used in the previous analyses [3, 22, 24]. This meant data for the four years 2012 to 2015 (i.e. beginning of 2012 to end of 2015) in Birmingham and Arlington, and four years in Bergen from 2016 to 2019. Despite the availability of data for additional years since the previous work, these were omitted for consistency and comparability with the previous work. For the two new locations, the data were for eight years in Leeds (2012 to 2019) and four years in Berlin (2016 to 2019), this representing the onset of data availability but stopping at the end of 2019 to avoid any confounds caused by including 2020, the year in which travel restrictions were imposed due to the Covid-19 pandemic. An additional criterion for the current analysis was that counter data were

**Table 3. Sources and extent of automated cycle count data.**

| Location | Number of counters | Date range | Data source |
|---|---|---|---|
| Arlington, VA, USA | 24 | 01/2012 to 12/2015 | https://counters.bikearlington.com/data-for-developers/ [35] |
| Bergen, Norway | 15 | 01/2016 to 12/2019 | https://trafikkdata.atlas.vegvesen.no/ [36] |
| Berlin, Germany | 26 | 01/2016 to 12/2019 | https://www.berlin.de/sen/uvk/mobilitaet-und-verkehr/verkehrsplanung/radverkehr/weitere-radinfrastruktur/zaehlstellen-und-fahrradbarometer/ [37] |
| Birmingham, UK | 43 | 01/2012 to 12/2015 | https://data.birmingham.gov.uk/dataset/cycling-sensors [38]* |
| Leeds, UK | 23 | 01/2012 to 12/2019 | https://datamillnorth.org/dataset/leeds-annual-cycle-growth- [39] |

* At the time of writing this database was not available online: we used a version previously downloaded.

retained only for complete years. This meant that for a counter installed (or removed) part-way through a year, that part-year of data were omitted, but subsequent complete years were retained (S1 Table shows the counters used in each year for each city).

The cycle counters in Arlington record inbound and outbound information separately at the same location: this analysis used the sum of inbound and outbound cyclists to determine the total number of cyclists, as was done previously [22].

## 2.2 Procedure

All possible darkness and daylight periods were defined according to data collected from the Time and Date website [40]. These data provide times of daily sunrise and sunset (solar altitude 0˚) and the transition from civil to nautical twilight (solar altitude -6˚). *Case* hours were those being in daylight (solar altitude >0˚) for one part of the year and in darkness (solar altitude <-6˚) for another part. We excluded dates where the case hours included any period in civil twilight. Note, however, that the data are reported at one minute intervals which means there could be inclusion of up to one minute of civil twilight. This criterion provides a more strict definition than used in previous studies where data collected in civil twilight were retained but included in either darkness [22] or the whole hour was allocated to the ambient light level phase present for the greater amount of time [3]. While the case period could be of any duration, from a few minutes to a few hours, we used hourly intervals because that matches the intervals at which cyclist count data are available.

Fig 1 shows the case and control hours for each city. Case hours where the daylight or darkness period was less than two months per year were excluded, targeting a larger sample and hence more accurate ORs for each set of case and control hour. This is a different approach to Wanvik [7], who included only those hours where there were at least 15 events in his analysis of road traffic crashes, but targets the same goal.

This led to the exclusion of four possible case hours: Arlington 6:00–06:59, Bergen 22:00–22:59, Berlin and Birmingham 05:00–05:59. Bergen does not have any dark control hours due to the short period between the start and end of civil twilight in summer. Arlington does not have any morning case hours due to the minimal difference between sunrise in winter and summer periods. S2 Table shows the case hours and daylight, twilight and darkness periods for each year and each city.

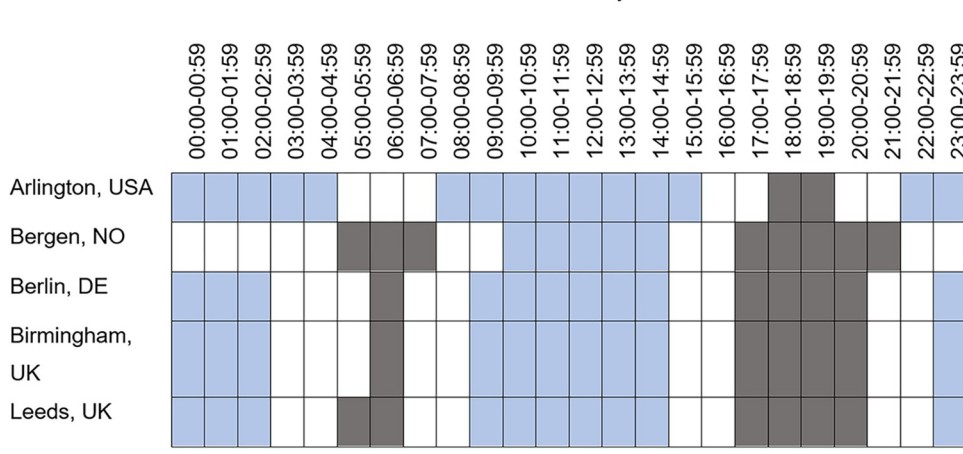

**Fig 1. Case and control hours included in the analyses.** Dark shaded cells are the case hours; light shaded cells are the control hours; unshaded cells were omitted.

To exclude civil twilight from the analysis a control hour had to be entirely in darkness with a solar altitude below -6˚ for the duration of the hour, or entirely in daylight with a solar altitude above 0˚ for the duration of the hour. This decision led to three changes with previous work. It precludes the control hour of 21:00 to 21:59 previously used in analysis of Arlington [22], the control hour of 22:00 to 22:59 previously used in analysis of Birmingham [3] and the control hour of 22:00 to 22:59 previously used in analysis of Bergen [24]. This is because for part of the year, these hours include civil twilight conditions. For example, for Arlington, 21:00–21:59 period is partially in twilight from June until July, and for Bergen, sunset is after 22:00 from May to August.

For each counter, the counts for each hour were checked for the presence of missing or negative values, indicating a fault in the counter operation. In those cases where such values were found, the numbers for that period were removed from the corresponding control or case hour. S3 Table shows the number of cyclists in day and darkness for each case and control hour.

The effect of ambient light level on cyclist numbers was revealed using the OR defined in Eq 1. This compares the numbers of cyclists in the daylight and dark periods of the case hour, with the numbers of cyclists in the control hour for the same times of year. An OR>1.0 shows that cyclist numbers are reduced in darkness, while OR = 1.0 shows that cyclist numbers were not affected by ambient light level. The OR was determined for all possible combinations of case hour and control hour at the specific location (Fig 1). For each OR, the 95% confidence interval (95%CIs) was calculated using Eq 2 [7]. The Holm-Bonferroni correction [41] was used to correct for multiple testing, with statistical significance indicated by an alpha level of 0.01. To determine the significance, the p-value was calculated using the method described by Sheskin [42].

$$R_{\text{odds}} = \frac{(A/B)}{(C/D)} \tag{1}$$

$$95\% \ CI = exp\left(Ln(R_{\text{odds}}) \pm 1.96 \ x\sqrt{\frac{1}{A} + \frac{1}{B} + \frac{1}{C} + \frac{1}{D}}\right) \tag{2}$$

Where
 $R_{\text{odds}}$ is the odds ratio (OR)
 A is the number of cyclists when the Case hour is in daylight
 B is the number of cyclists when the Case hour is in darkness
 C is the number of cyclists in the Control hour when the Case hour is in daylight
 D is the number of cyclists in the Control hour when the Case hour is in darkness

## 3 Results

Tables 4–8 show the ORs determined for each combination of case and control hour in Arlington, Bergen, Berlin, Birmingham and Leeds.

The ORs are significantly greater than 1.0 ($p<0.01$) in three cities; Arlington (29 out of 30 cases), Birmingham (all 50 cases) and Leeds (56 of 60 cases). The remaining cases in these three cities did not suggest a significant departure from 1.0. Considering only the significant cases:

- In Arlington the ORs ranged from 1.07 to 3.60. 25 exceeded the threshold for a small effect size, of which 11 exceeded the threshold for a medium effect including two which exceeded the threshold for a large effect.

**Table 4. Odds ratios for each combination of case hour and control hour in Arlington.** In all cases except one (18:00–18:59 case hour, 02:00–02:59 control hour) the OR is >1.0 ($p$<0.01).

| Control hours | | Case Hours | |
|---|---|---|---|
| | | **18:00–18:59** | **19:00–19:59** |
| Dark | 22:00–22:59 | 1.98 | 2.04 |
| | 23:00–23:59 | 1.44 | 1.66 |
| | 00:00–00:59 | 1.51 | 1.70 |
| | 01:00–01:59 | 1.22 | 1.36 |
| | 02:00–02:59 | 0.97 | 1.07 |
| | 03:00–03:59 | 1.17 | 1.23 |
| | 04:00–04:59 | 3.60 | 3.30 |
| Daylight | 08:00–08:59 | 1.62 | 2.00 |
| | 09:00–09:59 | 1.43 | 1.74 |
| | 10:00–10:59 | 1.22 | 1.45 |
| | 11:00–11:59 | 1.34 | 1.61 |
| | 12:00–12:59 | 1.59 | 1.94 |
| | 13:00–13:59 | 1.82 | 2.26 |
| | 14:00–14:59 | 1.98 | 2.52 |
| | 15:00–15:59 | 2.06 | 2.67 |

- In Birmingham the ORs ranged from 1.18 to 2.68: 46 exceeded the threshold for a small effect size and 18 a medium effect size.

- In Leeds the ORs ranged from 1.01 to 1.48: 29 exceeded the threshold for a small effect size.

In those three cities, Arlington, Birmingham and Leeds, the ORs show that darkness deters cycling. A different pattern exists for the two remaining cities, Bergen and Berlin. While the ORs for some combinations of case and control hour also suggest a deterrence effect of darkness on cycling, an OR>1.0, in other combinations the ORs are significantly smaller than 1.0.

For Bergen, 20 of the 40 cases show an OR>1.0 ($p$<0.01), ranging from 1.09 to 1.42, of which 15 exceed the threshold for a small effect size. The remaining 20 cases show an OR<1.0 ($p$<0.01), ranging from 0.71 to 0.96, of which ten exceed the threshold (0.82 [27]) for a small effect size.

For Berlin, 29 of the 50 cases show an OR>1.0 ($p$<0.01), ranging from 1.01 to 1.41, of which only nine exceed the threshold for a small effect size. 16 cases show an OR<1.0 ($p$<0.01), ranging from 0.77 to 0.98, of which three exceed the threshold for a small effect size. In the remaining five cases the OR does not depart from 1.0.

**Table 5. Odds ratios for each combination of case hour and control hour in Bergen.** In 20 out of 40 cases the OR is greater than 1.0 (p<0.01). In the remaining cases the OR<1.0 (p<0.01)–these are for case hours in the morning and at 17:00–17:59.

| Control hours | | Case hours | | | | | | | |
|---|---|---|---|---|---|---|---|---|---|
| | | **05:00–05:59** | **06:00–06:59** | **07:00–07:59** | **17:00–17:59** | **18:00–18:59** | **19:00–19:59** | **20:00–20:59** | **21:00–21:59** |
| Daylight | 10:00–10:59 | 0.87 | 0.76 | 0.79 | 0.96 | 1.16 | 1.35 | 1.41 | 1.32 |
| | 11:00–11:59 | 0.83 | 0.71 | 0.74 | 0.91 | 1.09 | 1.26 | 1.31 | 1.25 |
| | 12:00–12:59 | 0.84 | 0.71 | 0.72 | 0.91 | 1.09 | 1.25 | 1.31 | 1.24 |
| | 13:00–13:59 | 0.83 | 0.71 | 0.72 | 0.91 | 1.09 | 1.24 | 1.31 | 1.24 |
| | 14:00–14:59 | 0.89 | 0.78 | 0.78 | 0.96 | 1.17 | 1.34 | 1.42 | 1.35 |

**Table 6. Odds ratios for each combination of case hour and control hour in Berlin.** In 29 out of 50 cases (except the ones where OR is below 1.0, but also 06:00–06:59 case hour and 10:00–10:59, 13:00–13:59 control hours, 17:00–17:59 case hour and 01:00–01:59 control hour, and 20:00–20:59 case hour and 01:00–01:59 control hour) the OR is >1.0 (p<0.01). The ORs tend to be <1.0 for the morning case hour and for the dark control hours.

| Control hours | | Case hours | | | | |
|---|---|---|---|---|---|---|
| | | 06:00–06:59 | 17:00–17:59 | 18:00–18:59 | 19:00–19:59 | 20:00–20:59 |
| Dark | 23:00–23:59 | 0.78 | 0.95 | 0.93 | 0.93 | 0.98 |
| | 00:00–00:59 | 0.77 | 0.97 | 0.93 | 0.91 | 0.96 |
| | 01:00–01:59 | 0.80 | 1.00 | 0.94 | 0.93 | 1.00 |
| | 02:00–02:59 | 0.85 | 1.07 | 1.03 | 0.99 | 1.06 |
| Daylight | 09:00–09:59 | 1.07 | 1.22 | 1.25 | 1.30 | 1.41 |
| | 10:00–10:59 | 1.00 | 1.18 | 1.18 | 1.21 | 1.33 |
| | 11:00–11:59 | 0.98 | 1.14 | 1.15 | 1.18 | 1.32 |
| | 12:00–12:59 | 0.98 | 1.14 | 1.15 | 1.19 | 1.34 |
| | 13:00–13:59 | 1.00 | 1.16 | 1.16 | 1.21 | 1.36 |
| | 14:00–14:59 | 1.01 | 1.17 | 1.18 | 1.23 | 1.40 |

**Table 7. Odds ratios for each combination of case hour and control hour in Birmingham.** In all cases the OR is greater than 1.0 (p<0.01).

| Control hours | | Case hours | | | | |
|---|---|---|---|---|---|---|
| | | 06:00–06:59 | 17:00–17:59 | 18:00–18:59 | 19:00–19:59 | 20:00–20:59 |
| Dark | 23:00–23:59 | 1.41 | 1.61 | 1.66 | 2.06 | 2.26 |
| | 00:00–00:59 | 1.50 | 1.56 | 1.68 | 2.14 | 2.38 |
| | 01:00–01:59 | 1.67 | 1.85 | 1.91 | 2.35 | 2.57 |
| | 02:00–02:59 | 1.83 | 1.88 | 1.99 | 2.44 | 2.68 |
| Daylight | 09:00–09:59 | 1.26 | 1.37 | 1.46 | 1.90 | 2.14 |
| | 10:00–10:59 | 1.23 | 1.34 | 1.41 | 1.81 | 2.11 |
| | 11:00–11:59 | 1.21 | 1.31 | 1.37 | 1.76 | 2.05 |
| | 12:00–12:59 | 1.20 | 1.29 | 1.36 | 1.75 | 2.03 |
| | 13:00–13:59 | 1.20 | 1.30 | 1.35 | 1.75 | 2.01 |
| | 14:00–14:59 | 1.18 | 1.27 | 1.32 | 1.72 | 1.99 |

**Table 8. Odds ratios for each combination of case hour and control hour in Leeds.** In 56 out of 60 cases the OR is greater than 1.0 (p<0.01), the exceptions being 05:00–05:59 case hour with 01:00–01:59, 02:00–02:59, 11:00–11:59 and 23:00–23:59 control hours where the OR is not significantly different to 1.0.

| Control hours | | Case hours | | | | | |
|---|---|---|---|---|---|---|---|
| | | 05:00–05:59 | 06:00–06:59 | 17:00–17:59 | 18:00–18:59 | 19:00–19:59 | 20:00–20:59 |
| Dark | 23:00–23:59 | 1.02 | 1.24 | 1.23 | 1.26 | 1.32 | 1.31 |
| | 00:00–00:59 | 1.16 | 1.36 | 1.35 | 1.37 | 1.46 | 1.48 |
| | 01:00–01:59 | 1.05 | 1.27 | 1.25 | 1.27 | 1.38 | 1.37 |
| | 02:00–02:59 | 1.05 | 1.30 | 1.35 | 1.30 | 1.35 | 1.38 |
| Daylight | 09:00–09:59 | 1.11 | 1.22 | 1.15 | 1.22 | 1.36 | 1.42 |
| | 10:00–10:59 | 1.07 | 1.14 | 1.09 | 1.15 | 1.27 | 1.35 |
| | 11:00–11:59 | 1.01 | 1.08 | 1.05 | 1.10 | 1.20 | 1.29 |
| | 12:00–12:59 | 1.04 | 1.11 | 1.05 | 1.10 | 1.21 | 1.31 |
| | 13:00–13:59 | 1.04 | 1.13 | 1.07 | 1.12 | 1.23 | 1.31 |
| | 14:00–14:59 | 1.03 | 1.12 | 1.07 | 1.11 | 1.22 | 1.31 |

## 4 Discussion

### 4.1 Changes in the OR

The results suggest a deterrence effect of darkness on cycling in Arlington, Birmingham and Leeds, as indicated by ORs significantly greater than 1.0 and tending to suggest a small effect size or greater. In contrast, mixed results were found for Berlin and Bergen, with darkness tending to induce a deterrence effect when using evening case hours and daylight control hours, but suggesting the opposite when using morning case hours and dark control hours. In further work [43] we propose that low ORs in morning case hours can be explained through variations in the relative proportions of utilitarian and recreational cyclist journeys at different times of day. Here we are concerned primarily with the choice of different case and control hours.

Fig 2 shows ORs plotted against control hour, plotted separately for each case hour, for Bergen, Berlin, Birmingham and Leeds. The changes in control hour lead to progressive changes

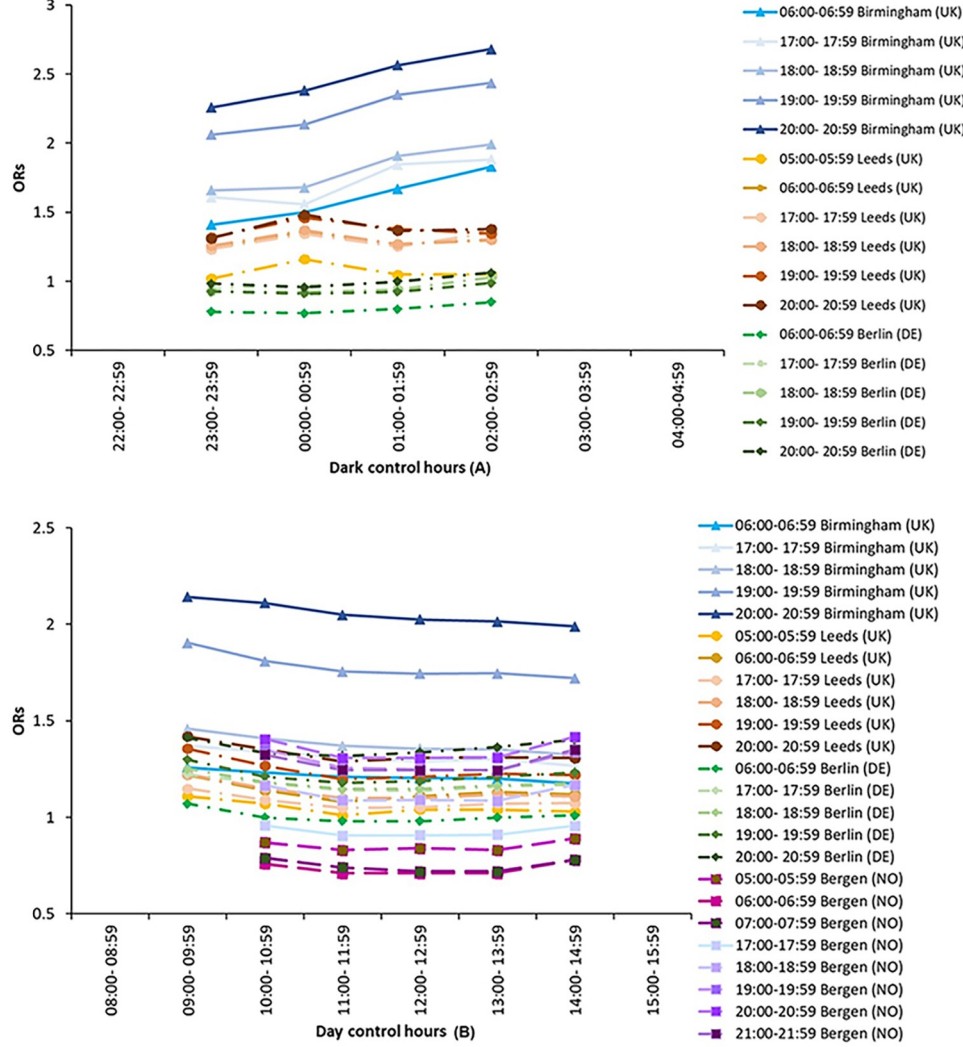

**Fig 2.** Changes in OR with control hour for Bergen, Berlin, Birmingham and Leeds: (A) *dark* control hours, (B) *daylight* control hours. Each line represents a different combination of case hour and city as shown in legend.

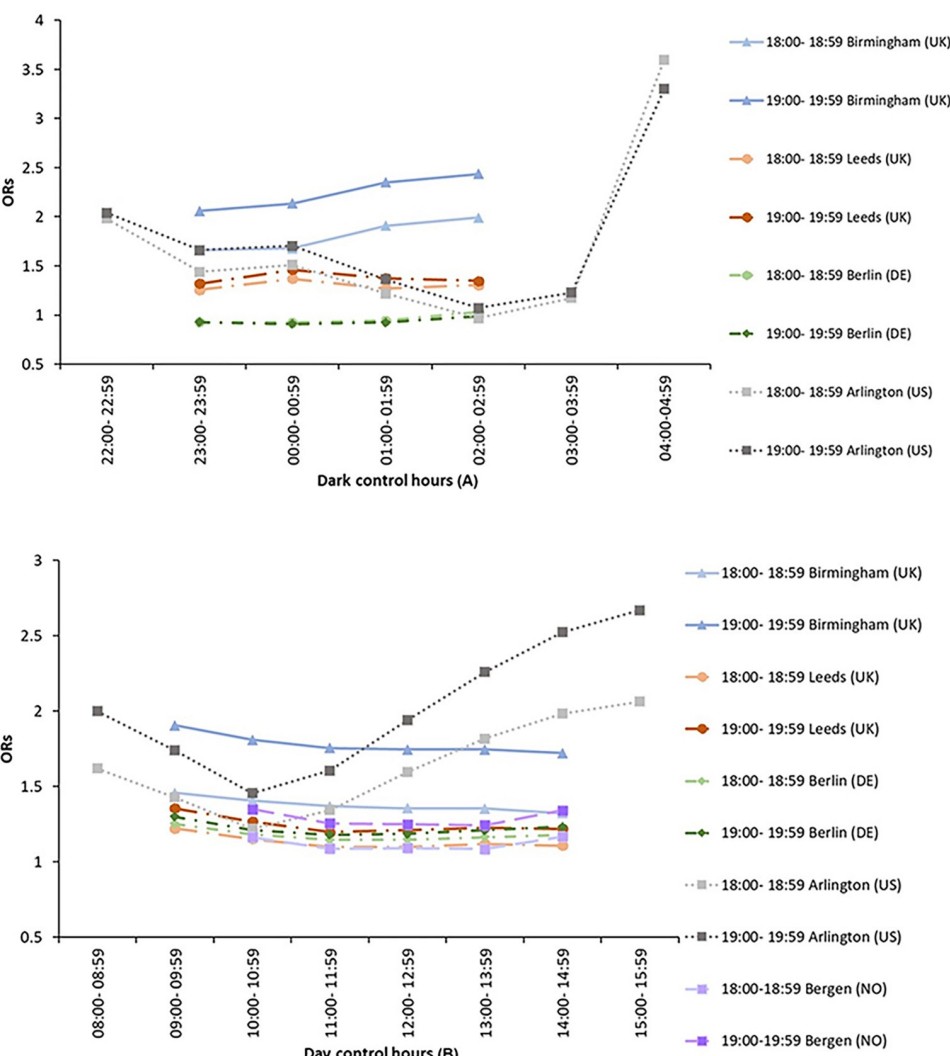

**Fig 3.** Changes in OR with control hour for Arlington, Bergen, Berlin, Birmingham and Leeds: (A) *dark* control hours, (B) *daylight* control hours. Each line represents a different combination of case hour and city as shown in legend.

in OR, meaning that the trend lines display smooth variations. For Bergen, Berlin and Leeds there is little change in OR with case hour, but for Birmingham the ORs progressively increase for dark control hours and decrease for daylight control hours. That suggests little effect of the choice of control hour because a similar OR is estimated when a different choice of control hour is made.

For Arlington, however, the change in OR with change in control hour does not follow such a progressive trend. Fig 3 shows ORs plotted against control hour, for Arlington and for the matching case hours from Bergen, Berlin, Birmingham and Leeds. For the dark control hours, ORs decrease from about 2.0 to 1.2 from 22:00 to 04:00, but then reverses to an OR of over 3.0 at 04:00. For the daylight control hours a similar trend is seen, with ORs for control hours at 08:00 first decreasing and then increasing. For Arlington the choice of control hour is more significant because a change in that choice could lead to a large change in the resultant OR.

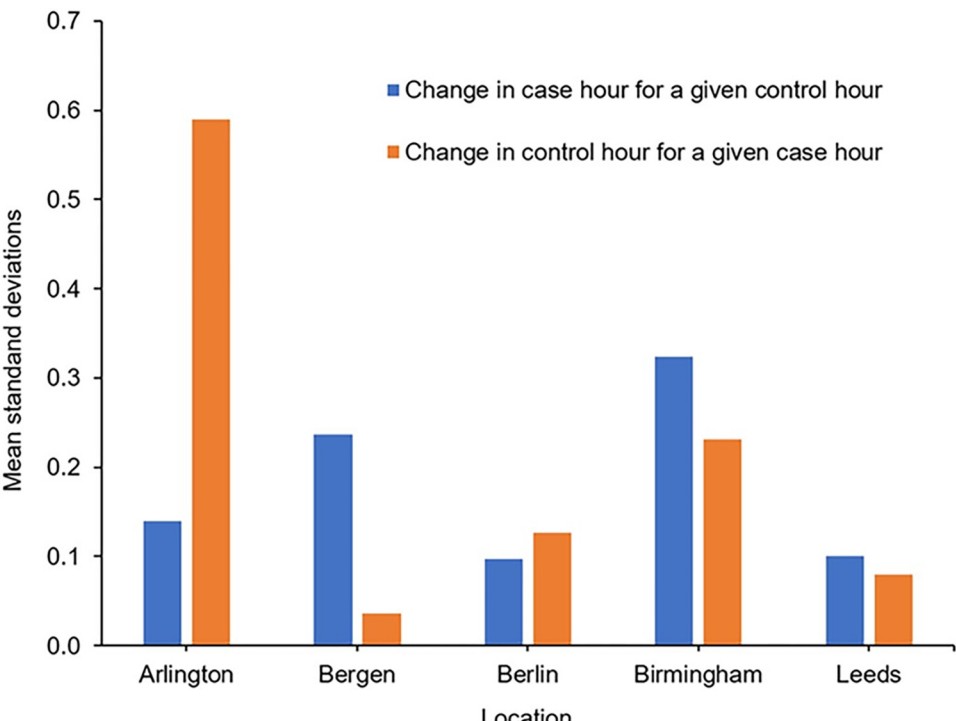

**Fig 4. Mean standard deviation of the odds ratio when either the case or control hour is held constant.**

Fig 4 shows the mean standard deviation about the odds ratio when the case hour is held constant and the control hour is varied (and vice versa). Consider for example the control hour of 11:00 to 11:59 in Birmingham: across the five case hours the OR ranges from 1.21 to 2.05, with a population standard deviation of 0.316. There are ten control hours in Birmingham, these having standard deviations ranging from 0.307 to 0.347, across which the mean standard deviation is 0.324. For Arlington, change in control hour for a given case hour leads to a much higher standard deviation than does change in case hour for a given control hour. For Bergen the opposite trend is found. There is no consistent trend for the choice of one (i.e. choice of case or control hour) to have a smaller or greater effect than the choice of the other.

## 4.2 Weighted mean OR

A matrix of ORs such as shown in Tables 4 to 8 is not readily legible or easy to interpret. As with reporting of the colour rendition or colour fidelity qualities of a light source [44, 45] it may be useful to reduce these data to a single figure. This approach of reducing multiple odds ratios to a single summary figure is also done in meta-analyses when calculating a pooled odds ratio as a measure of the overall effect across multiple studies [46].

When analysing count data for Arlington using instead the whole year method with two control hours, Fotios et al [22] found ORs of 1.75 and 1.22 for the control hours starting at 15:00 (daylight) and 21:00 (dark) respectively. The arithmetic mean (1.49) of these two ORs does not match the OR determined directly using the combined data (1.67). This may arise where there are significant differences in the numbers of cyclists at different hours, which would influence the OR determined using the two control hours simultaneously, but would not influence the average of ORs determined using the two control hours separately. We therefore considered a weighted mean, with the weighting being the numbers of cyclists in the case

**Table 9. The weighted mean ORs.**

| OR characteristic | | Counter location | | | | |
|---|---|---|---|---|---|---|
| | | Arlington | Bergen | Berlin | Birmingham | Leeds |
| $MH_{\text{w-odds}}$ | | **1.80** | **1.01** | **1.16** | **1.56** | **1.18** |
| 95% CI | Lower | 1.80 | 1.01 | 1.16 | 1.56 | 1.18 |
| | upper | 1.81 | 1.02 | 1.16 | 1.57 | 1.18 |
| *p*-value | | <0.001 | <0.001 | <0.001 | <0.001 | <0.001 |
| Effect size | | Small | Negligible | Negligible | Small | Negligible |

and control hours for each specific OR, as shown in Eq 3. This follows the Mantel-Haenszel method for calculating pooled odds ratios, originally developed for assessment of factors associated with disease [47, 48]. 95% Confidence Intervals for the weighted OR were calculated using the Robins, Breslow and Greenland method (*V*) as shown in Eqs 4 and 5 [48].

$$MH_{\text{w-odds}} = \frac{\sum \frac{AD}{N}}{\sum \frac{BC}{N}} \qquad (3)$$

Where

$MH_{\text{w-odds}}$ is Mantel-Haenszel pooled OR

$N$ = total number of cyclists in the case and control hours (A+B+C+D)

$A$, $B$, $C$, $D$ are defined as for Eqs 1 and 2

$$95\% \, CI = exp(ln(MH_{\text{w-odds}}) \pm 1.96 \, x \, \sqrt{V}) \qquad (4)$$

where

$$V = \frac{\sum RP}{2R^2} + \frac{\sum (PS + QR)}{2RS} + \frac{\sum SQ}{2S^2} \qquad (5)$$

Where,

$P$ = (A+D) / N

$Q$ = (B+C) / N

$R$ = (AD) / N

$S$ = (BC) / N

Table 9 shows the weighted mean OR for each city. In each case the OR is suggested to be significantly greater than 1.0, indicating a relative reduction in cycling after dark, with these ORs suggesting a negligible effect in three cities (Bergen, Berlin and Leeds) and exceeding the threshold for a small effect size in two cities (Arlington and Birmingham). In further work we show how these effect sizes vary between weekdays and weekends (a proxy for type of cyclist journey), with effect sizes tending to increase in particular at weekends, and also vary for different counter locations [43, 49].

## 4.3 Comparison with previous studies

Previous analyses of cycling flows in Arlington reported overall ORs of 1.38 [17] using the clock change method and 1.67 [22] using the whole year method. The current study revealed a weighted mean OR of 1.80. While this is a slightly higher estimate, it retains the statistically significant effect and a small effect size of those previous estimates.

Previous analyses of cycling flows in Birmingham reported an overall OR of 1.32 [3] using the whole year method which is lower than the weighted mean of 1.56 found in the current

work, but again retains the conclusion of an effect which is statistically and substantively significant.

In Arlington and Birmingham, the ORs were significantly greater than 1.0 in all but one of the case and control hour combinations reported here (Tables 4 and 7). For these two cities, using a single combination of case and control hour would be unlikely to lead to an incorrect conclusion about the deterrent effect of darkness, albeit there being some variance in the OR. A similar conclusion would be drawn for Leeds.

For Bergen and Berlin, however, the choice of a specific case and control hour would have an impact on the conclusion drawn about the effect of darkness on cycling rates, with some combinations suggesting a significant reduction in cycling, some suggesting a significant increase in cycling, and others suggesting no effect. For these two cities the choice of case and control hour therefore has greater influence on the outcome, and hence we suggest instead to use the weighted mean OR from all possible combinations of case and control hour.

## 4.4 Limitations

A first limitation to this analysis is that all five locations are for populations which are western, industrialised, educated and with access to multiple travel mode options. Analyses of cities representing other contexts may show different trends. We are therefore analysing a separate and larger number of locations in a parallel project.

A second limitation is that this analysis uses data from counters installed by others and hence reasons for the choice of locations are not known. This may bias the overall number of cyclists counted and the proportions of utilitarian and recreational cyclist journeys in that sample. We are investigating this using an alternative source of data, this being crowd-sourced cyclist journeys captured using the trip-recording app STRAVA.

A third limitation is that the underlying data need to be further refined to more clearly reveal where a significant and substantive deterrent effect of darkness on cycling prevails. The variation in ORs might be attributed to several factors, such as differences in cycling culture, changes in lighting policies, location of the counters (e.g., major or minor roads) or variations in reasons to cycle (e.g., recreational vs utilitarian). A tentative analysis suggests that distance from the city centre and road type (whether major or minor) explains some of the variance [49]. In further work we report on the effects of type of journey [43].

## 5 Conclusion

This study was conducted to examine the impact of darkness on the numbers of people cycling, using an odds ratio to establish changes in cycling across different case and control periods. Specifically, the study investigated the extent to which the choice of case and control hour effects the OR. Cyclist count data from automated counters in five cities were included–Arlington, Bergen, Berlin, Birmingham and Leeds.

The results show that darkness tends to have a deterrent effect on cycling, revealed by ORs significantly greater than 1.0.

Within each city, ORs were determined for all possible combinations of case and control hour. The results show variance across these combinations: ORs vary with the choice of case and control hour, and hence also the statistical and substantive significance of the effect. For Birmingham and Leeds this variance was small, suggesting that the precise choice of case and control hour would have little impact on the outcome, with each combination of case and control hour tending to suggest a significant reduction in cycling after dark. For Arlington the change in OR was larger but was always in the same direction suggesting a deterrent effect of darkness. For Arlington, therefore, the choice of case and control hour is more critical because

a change in that decision could lead to a large change in the resultant OR. For the remaining two cities, Bergen and Berlin, there was a large variance in ORs, with different conclusions about the deterrent effect of darkness being indicated for different combinations of case and control hour–some suggesting either a significant reduction in cycling after dark, a significant increase, or no effect.

Given that the choice of case and control hour can have an effect, we suggest that future work considers all possible combinations of case and control hour, and uses a weighted mean OR to establish a best estimate of the effect. Here we used the Mantel-Haenszel pooled odds ratio, which is essentially the arithmetic mean weighted by the numbers of cyclists.

This work confirms the conclusions of previous work (Table 2) that had used only a limited choice of case and control hours in demonstrating that darkness has a deterrent effect on cycling.

Evidence that darkness has a deterrent effect on cycling provides support for the use of road lighting after dark to encourage people to cycle rather than to use motorised transport or to avoid travelling. This in turn supports global initiatives to promote active travel and this improve the sustainability of travel.

## Supporting information

**S1 Table. Counter information for each city.**
(CSV)

**S2 Table. Case and control hours used for each city.**
(CSV)

**S3 Table. The number of cyclists for each case and control hour.**
(CSV)

## Acknowledgments

The authors thank the local authorities that made their data publicly available.

## Author Contributions

**Conceptualization:** Steve Fotios, Jim Uttley.

**Data curation:** Demet Yesiltepe.

**Formal analysis:** Demet Yesiltepe, Maan Balela.

**Funding acquisition:** Steve Fotios, Jim Uttley.

**Investigation:** Demet Yesiltepe, Maan Balela.

**Methodology:** Steve Fotios, Jim Uttley, Demet Yesiltepe, Maan Balela.

**Project administration:** Steve Fotios, Jim Uttley.

**Supervision:** Steve Fotios, Jim Uttley.

**Validation:** Demet Yesiltepe.

**Visualization:** Demet Yesiltepe.

**Writing – original draft:** Steve Fotios, Jim Uttley, Demet Yesiltepe, Maan Balela.

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
