## [Decision Letter · Decision Letter 0]

18 Jul 2024

PONE-D-24-14676Variance in odds ratios for estimating the deterrent effect of darkness on cycling: Variation due to the choice of case and control hoursPLOS ONE

Dear Dr. Yesiltepe,

Thank you for submitting your manuscript to PLOS ONE. After careful consideration, we feel that it has merit but does not fully meet PLOS ONE’s publication criteria as it currently stands. Therefore, we invite you to submit a revised version of the manuscript that addresses the points raised during the review process.

Furthermore, be sure you have attended the following points:

• Please make sure that the structure for citing published literature in the text, as well as the style of references in the References section, are consistent with the journal's style (see Instructions to Authors).

• English language needs revision for style and syntax.

• Abstract must be rewritten. I suggest focusing the abstract on your study and your results.

• Please add the originality of the study and add hypothesis at the end of the introduction section. Be please be more specific.

• Include more characteristics of participants. More information about the participant’s selection needed.

• Results are not clear. I would only appreciate to read a detailed statistical approach

• Please discuss the results of the study in relation to the previous studies.

• Add the public implications of this study.

We look forward to receiving your revised manuscript.

Kind regards,

Flávio Oliveira Pires, PhD

Academic Editor

PLOS ONE

Journal Requirements:

"This work was conducted within SATURN (Supporting Active Travel Using Road-lighting at Night), funded by the Engineering and Physical Sciences Research Council (EP/V043587/1)."

4. Please ensure that you refer to Figure 4 in your text as, if accepted, production will need this reference to link the reader to the figure.

Reviewers' comments:

Reviewer's Responses to Questions

**Comments to the Author**

1. Is the manuscript technically sound, and do the data support the conclusions?

Reviewer #1: Yes

2. Has the statistical analysis been performed appropriately and rigorously? 

Reviewer #1: Yes

3. Have the authors made all data underlying the findings in their manuscript fully available?

Reviewer #1: Yes

4. Is the manuscript presented in an intelligible fashion and written in standard English?

Reviewer #1: Yes

5. Review Comments to the Author

Reviewer #1: Thank you to the editor and the authors for the opportunity to review this paper entitled “Variance in odds ratios for estimating the deterrent effect of darkness on cycling: Variation due to the choice of case and control hours. This study investigated the impact of the choice of case and control hours on ORs obtained for cyclist flows in different countries.

The idea of this paper is interesting, and it seems to be important to help the knowledge of the area. Thus, below are some points that, in my opinion, would help the authors increase the quality of the study.

ABSTRACT

In line 17 is the first moment the word “odds ratios” appears, but there is no acronym description, OR, however, in line 20, the acronym ORs appears. So, please, insert the acronym in the first description of the odds ratios.

Additionally, it would be important, that the authors insert the “p values” in the abstract, rather than only describe “…the variance was larger… (line 21)”, or “…this suggested a statistically significant deterrent… (line 25)”.

INTRODUCTION

I suggest the authors remove or reorder (ex. supplementary document or only citing the study) both tables 1 and 2 from the introduction. The way they were inserted affects the fluidity of reading.

METHODS

Line 175. There is a double “that” written.

Why did the authors use eight years in the new locations rather than four years? And, why choose different years? Could these differences in the years impair the analysis?

DISCUSSION

Why were the formulas inserted in the discussion?

The discussion highlights the study's important findings, pertinent conclusions, and their implications or applications. So, it would be better to transfer the formulas for the Method session.

While in the introduction, the authors talk about the influence of the weather on the flow of cyclists, in the discussion, the topic is not presented. I think that the authors would invest in this topic.

6. PLOS authors have the option to publish the peer review history of their article (what does this mean?). If published, this will include your full peer review and any attached files.

Reviewer #1: No

---

## [Author Response · Author response to Decision Letter 0]

14 Aug 2024

A response letter is provided to answer both the reviewer's and editor's comments.

---

## [Decision Letter · Decision Letter 1]

17 Sep 2024

PONE-D-24-14676R1Variance in odds ratios for estimating the deterrent effect of darkness on cycling: Variation due to the choice of case and control hoursPLOS ONE

Dear Dr. Yesiltepe,

Thank you for submitting your manuscript to PLOS ONE. After careful consideration, we feel that it has merit but does not fully meet PLOS ONE’s publication criteria as it currently stands. Therefore, we invite you to submit a revised version of the manuscript that addresses the points raised during the review process.

**ACADEMIC EDITOR: **Congratulations, the article is close to the acceptance.

Please, addresses the recommnedation raised by the reviewer on item 7 (regarding the equations cited in the discussion). Please, remove the equations to "methods section" if they were used in the study's procedures or to a more suitable place (tables or supplementary material if applicable).

We look forward to receiving your revised manuscript.

Kind regards,

Flávio Oliveira Pires, PhD

Academic Editor

PLOS ONE

Journal Requirements:

Additional Editor Comments:

Dear authors,

Congratulations, the article is close to the acceptance.

Please, addresses the recommnedation raised by the reviewer on item 7 (regarding the equations cited in the discussion). Please, remove the equations to "methods section" if they were used in the study's procedures or to a more suitable place (tables or supplementary material if applicable).

Reviewers' comments:

Reviewer's Responses to Questions

**Comments to the Author**

1. If the authors have adequately addressed your comments raised in a previous round of review and you feel that this manuscript is now acceptable for publication, you may indicate that here to bypass the “Comments to the Author” section, enter your conflict of interest statement in the “Confidential to Editor” section, and submit your "Accept" recommendation.

Reviewer #1: All comments have been addressed

2. Is the manuscript technically sound, and do the data support the conclusions?

Reviewer #1: Yes

3. Has the statistical analysis been performed appropriately and rigorously? 

Reviewer #1: Yes

4. Have the authors made all data underlying the findings in their manuscript fully available?

Reviewer #1: Yes

5. Is the manuscript presented in an intelligible fashion and written in standard English?

Reviewer #1: Yes

6. Review Comments to the Author

Reviewer #1: Thank you to the editor and the authors for the opportunity to review again this paper entitled “Variance in odds ratios for estimating the deterrent effect of darkness on cycling: Variation

due to the choice of case and control hours.

The authors clarified all issues and made the changes to the manuscript when was relevant.

7. PLOS authors have the option to publish the peer review history of their article (what does this mean?). If published, this will include your full peer review and any attached files.

Reviewer #1: No

---

## [Author Response · Author response to Decision Letter 1]

26 Sep 2024

In the methods section, we present two equations (1 and 2) used to determine odds ratios (OR) and associated confidence intervals. This is done for each combination of case and control hours, leading to five matrices of ORs (Tables 4 to 8). It is only after detecting the variance in each matrix that the need for a weighted mean becomes apparent, and the equations (3, 4, and 5) necessary for that are thus presented after the results, in the discussion section.

We would prefer to keep equations 3 to 5 in the discussion section, as this better maintains the flow of information. Therefore, no changes have been made to the document.

---

## [Editor Report · Decision Letter 2]

29 Sep 2024

Variance in odds ratios for estimating the deterrent effect of darkness on cycling: Variation due to the choice of case and control hours

PONE-D-24-14676R2

Dear Dr. Yesiltepe,

We’re pleased to inform you that your manuscript has been judged scientifically suitable for publication and will be formally accepted for publication once it meets all outstanding technical requirements.

Kind regards,

Flávio Oliveira Pires, PhD

Academic Editor

PLOS ONE
---

## [Editor Report · Acceptance letter]

2 Oct 2024

PONE-D-24-14676R2 

PLOS ONE

Dear Dr. Yesiltepe, 

I'm pleased to inform you that your manuscript has been deemed suitable for publication in PLOS ONE. Congratulations! Your manuscript is now being handed over to our production team.

Kind regards, 

on behalf of

BSc PhD Flávio Oliveira Pires 

Academic Editor

PLOS ONE